# Seroprevalence of SARS-CoV-2 IgG Antibodies and Factors Associated with SARS-CoV-2 IgG Neutralizing Activity among Primary Health Care Workers 6 Months after Vaccination Rollout in France

**DOI:** 10.3390/v14050957

**Published:** 2022-05-03

**Authors:** Dorine Decarreaux, Marie Pouquet, Cecile Souty, Ana-Maria Vilcu, Pol Prévot-Monsacre, Toscane Fourié, Paola Mariela Saba Villarroel, Stephane Priet, Hélène Blanché, Jean-Marc Sebaoun, Jean-François Deleuze, Clément Turbelin, Andréas Werner, Fabienne Kochert, Brigitte Grosgogeat, Pascaline Rabiega, Julien Laupie, Nathalie Abraham, Caroline Guerrisi, Harold Noël, Sylvie van der Werf, Fabrice Carrat, Thomas Hanslik, Remi Charrel, Xavier de Lamballerie, Thierry Blanchon, Alessandra Falchi

**Affiliations:** 1Laboratoire de Virologie, Université de Corse Pascal Paoli, UR7310 Bioscope, 20250 Corte, France; 2INSERM, Institut Pierre Louis d’Epidémiologie et de Santé Publique, (IPLESP), Sorbonne Université, 75012 Paris, France; cecile.souty@iplesp.upmc.fr (C.S.); ana-maria.vilcu@iplesp.upmc.fr (A.-M.V.); pol.prevost@iplesp.upmc.fr (P.P.-M.); clement.turbelin@iplesp.upmc.fr (C.T.); caroline.guerrisi@iplesp.upmc.fr (C.G.); fabrice.carrat@iplesp.upmc.fr (F.C.); thomas.hanslik@aphp.fr (T.H.); thierry.blanchon@iplesp.upmc.fr (T.B.); 3Unité des Virus Emergents, Aix Marseille University, IRD 190, INSERM U1207, 13005 Marseille, France; toscane.fourie@gmail.com (T.F.); marielasaba@gmail.com (P.M.S.V.); stephpriet@gmail.com (S.P.); remi.charrel@univ-amu.fr (R.C.); xavier.de-lamballerie@univ-amu.fr (X.d.L.); 4Fondation Jean Dausset-CEPH, 75000 Paris, France; helene.blanche@fjd-ceph.org (H.B.); jean-marc.sebaoun@fjd-ceph.org (J.-M.S.); jean-françois.deleuze@fjd-ceph.org (J.-F.D.); 5Association Française de Pédiatrie Ambulatoire (AFPA), 69000 Orléans, France; docteur.werner.pediatre@wanadoo.fr (A.W.); fabienne.kochert@wanadoo.fr (F.K.); 6Faculté d’Odontologie, Université Claude Bernard Lyon 1, Université de Lyon, 69000 Lyon, France; brigitte.grosgogeat@univ-lyon1.fr; 7Laboratoire des Multimatériaux et Interfaces, UMR CNRS 5615, Université Claude Bernard Lyon 1, Université de Lyon, 69000 Lyon, France; 8Réseau ReCOL, Association Dentaire Française, 75000 Paris, France; reseau.recol@gmail.com; 9Service d’Odontologie, Hospices Civils de Lyon, 69007 Lyon, France; 10IQVIA, Réseau de Pharmaciens, 75000 Paris, France; pascaline.rabiega@iqvia.com (P.R.); nabraham@fr.imshealth.com (N.A.); 11Infectious Diseases Division, Santé Publique France, 94410 Saint Maurice, France; harold.noel@santepubliquefrance.fr; 12Institut Pasteur, Université Paris Cité, CNRS UMR3569, Molecular Genetics of RNA Viruses Unit, 75015 Paris, France; sylvie.van-der-werf@pasteur.fr; 13Institut Pasteur, Université Paris Cité, National Reference Center for Respiratory Viruses, 75015 Paris, France; 14Département de Santé Publique, Hôpital Saint-Antoine, APHP, 75012 Paris, France; 15Faculty of Health Sciences Simone Veil, Université de Versailles Saint-Quentin-en-Yvelines, UVSQ, UFR de Médecine, 78000 Versailles, France; 16Service de Médecine Interne, Hôpital Ambroise Paré, Assistance Publique—Hôpitaux de Paris (APHP), 92100 Boulogne Billancourt, France

**Keywords:** seroprevalence, SARS-CoV-2 antibodies, neutralizing antibodies, primary care, health care workers

## Abstract

We aimed to investigate the immunoglobulin G response and neutralizing activity against severe acute respiratory syndrome coronavirus-2 (SARS-CoV-2) among primary health care workers (PHCW) in France and assess the association between the neutralizing activity and several factors, including the coronavirus disease 2019 (COVID-19) vaccination scheme. A cross-sectional survey was conducted between 10 May 2021 and 31 August 2021. Participants underwent capillary blood sampling and completed a questionnaire. Sera were tested for the presence of antibodies against the nucleocapsid (N) protein and the S-1 portion of the spike (S) protein and neutralizing antibodies. In total, 1612 PHCW were included. The overall seroprevalences were: 23.6% (95% confidence interval (CI) 21.6–25.7%) for antibodies against the N protein, 94.7% (93.6–95.7%) for antibodies against the S protein, and 81.3% (79.4–83.2%) for neutralizing antibodies. Multivariate regression analyses showed that detection of neutralizing antibodies was significantly more likely in PHCW with previous SARS-CoV-2 infection than in those with no such history among the unvaccinated (odds ratio (OR) 16.57, 95% CI 5.96–59.36) and those vaccinated with one vaccine dose (OR 41.66, 95% CI 16.05–120.78). Among PHCW vaccinated with two vaccine doses, the detection of neutralizing antibodies was not significantly associated with previous SARS-CoV-2 infection (OR 1.31, 95% CI 0.86–2.07), but was more likely in those that received their second vaccine dose within the three months before study entry than in those vaccinated more than three months earlier (OR 5.28, 95% CI 3.51–8.23). This study highlights that previous SARS-CoV-2 infection and the time since vaccination should be considered when planning booster doses and the design of COVID-19 vaccine strategies.

## 1. Introduction

Since the identification of severe acute respiratory syndrome coronavirus-2 (SARS-CoV-2) as the etiological agent of coronavirus disease 2019 (COVID-19), efforts have been made worldwide to prevent infection and disease [1,2,3,4]. Since December 2020, highly effective vaccines that offer high levels of protection against severe disease have been introduced worldwide. However, they do not confer mucosal immunity preventing virus entry into the respiratory tract [1,2,3,4]. In France, national vaccination campaigns were rapidly implemented at the end of 2020 to limit hospitalizations, severe consequences, and deaths due to COVID-19 [5,6]. The initial campaign phase targeted populations at risk of severe COVID-19, but it was also rapidly implemented for health care workers (HCWs) aged > 50 years on 2 January 2021, and then for all voluntary HCWs on 6 February 2021. The vaccination schedule consisted of two vaccine doses, except for the vaccination of people who had recovered from COVID-19 within 6 months before vaccination [5,6]. When mostly mild cases of breakthrough infection were reported in different countries, there was a concern that individuals in whom the immune response to the standard vaccination schedule rapidly waned could become seriously ill if infected [7,8,9]. The waning of vaccine-induced immunity over time and a possible increased immune evasion by the SARS-CoV-2 variants led many countries, including France, to plan the administration of a booster dose of the COVID-19 vaccine from September 2021 [5,10]. The first variants of concern (VOCs) appeared at the end of 2020. The Alpha variant spread rapidly in France after its introduction at the end of 2020 and became the majority variant in March 2021. The Beta and Gamma variants also circulated in the first half of 2021, although to a lesser extent. The Delta variant appeared in May 2021 and rapidly replaced the previous variants, becoming the major variant in France in July 2021 and represented more than 99% of the circulating variants from August 2021 [11]. 

More than two years after the beginning of the pandemic, the main priority of the scientific community remains the evaluation of the immunogenicity of the vaccines. Field data on vaccine immunogenicity are important for monitoring the performance of COVID-19 vaccination programs and for the planning of public health policy decisions. Various antigens have been studied for this purpose, but the nucleocapsid (N) protein and spike (S) protein are the most widely studied SARS-CoV-2 antigens. A proportion of the spike-binding antibodies are likely to have neutralizing activity. Thus, numerous studies have shown a correlation between S protein binding assays and various forms of functional virus neutralization assays [12]. Although it has been suggested that neutralizing antibodies are likely to be highly important for efficient protection against reinfection and could serve as a correlate for vaccine protection against SARS-CoV-2 in humans [13], few studies have relied on neutralizing antibodies [14].

Primary health care workers are at risk of COVID-19 through work and community exposure as well as being potential transmission sources of nosocomial infection for patients and co-workers [15]. Thus, six months after the COVID-19 vaccination campaign was launched, we conducted a French nationwide cross-sectional survey to investigate the immunoglobulin (Ig) G response and neutralizing activity in a sample of primary HCW (PHCW) populations to evaluate the titer and duration of antibody responses elicited by each vaccination scheme. For this purpose, we considered the number of COVID-19 doses received, previous SARS-CoV-2 infection, demographic factors, and the time since vaccination for the vaccinees. 

## 2. Materials and Methods

### 2.1. Study Design, Time Frame, and Population

We conducted a cross-sectional study between 10 May 2021 and 31 August 2021, among the following four PHCW populations throughout metropolitan France: general practitioners (GPs), pediatricians, pharmacists and assistants, and dentists and assistants. In France, HCWs aged >50 years were eligible for COVID-19 vaccination from 4 January 2021, and all of them, regardless of age, became eligible on February 5, 2021 [16]. At the time of the survey, the third COVID-19 wave in France had ended (March 2021–May 2021). This study followed the Strengthening the Reporting of Observational Studies in Epidemiology (STROBE) reporting guideline for cross-sectional studies.

### 2.2. Participant Sampling

The protocol study has been described in detail elsewhere [16]. Briefly, the PHCWs were recruited from the following four primary care research and monitoring networks: the French *Sentinelles* Network (GPs), the French Association of Ambulatory Pediatrics (pediatricians), the ReCOL network (dentists and assistants), and IQVIA (pharmacists and assistants). The PHCWs were invited by each network to enroll in the study via e-mail communication, virtual meetings, and announcements on the social media platforms for each network. All PHCWs were eligible to participate, except for those who had participated in a chemoprophylaxis clinical trial of SARS-CoV-2 infection. Participation was voluntary, and the participants registered online. Electronic informed consent was obtained from each participant before their enrolment in the study.

### 2.3. Data Collection and Sample Preparation 

Data collection included capillary blood sampling and a questionnaire. 

Collection kits were mailed to each participant for at-home sample collection, including a dried blood spot (DBS) card, lancets, a pad, detailed printed instructions for performing the blood sampling, and a prepaid envelope. Returned DBSs were received at a centralized biobank (CEPH Biobank, Paris, France), and blood spots were visually assessed, registered, and punched in 2D FluidX 96-Format 0.5-mL tubes (Brooks, Chelmsford, MA, USA). Tubes containing the punches were stored before shipment at room temperature and were sent to a virology laboratory (Unité des Virus Emergents, Marseille, France) for serological analysis.

At the time of specimen collection (from May to August 2021), the PHCWs were also asked to complete a computer-based questionnaire about demographic characteristics, occupation, and clinical information, including the history since January 2020 of SARS-CoV-2 infection and COVID-19 vaccination.

### 2.4. Laboratory Analysis

Four chads from each dried blood spot were diluted in 380 µL PBS with 1% of penicillin/streptomycin using EpMotion 96 in fluidX micro tubes. The tubes were shaken in 96 well plates at 1050 rpm for 30 min followed by centrifugation at 4500× *g* for 10 min. The volume of PBS, which was decided after obtaining correspondence results for more than 1000 serum and dried blood spots, corresponds to a titration of 40. Moreover, before performing the ELISA, the hemoglobin levels in the blood spots were quantified to evaluate the quality of the serum obtained from each blood spot. The threshold median hemoglobin level was decided after more than 1000 samples were tested beforehand. The ELISA and virus-neutralizing tests were performed using sera obtained from blood spots.

All samples were tested for antibodies against the N protein (NTD and CTD domains) and the S-1 portion of the S protein as well as neutralizing activity against SARS-CoV-2. The NTD and CTD domains of the SARS-CoV-2 N proteins used in the Luminex assay were available at the European Viral Archive goes global (EVAg) repository (www.european-virus-archive.com (accessed on 14 February 2022), ref: 100P-03956 and 100P-03957). Optimal antigen quantities and sample dilutions were selected based on the highest signal-to-noise ratio, most reproducible results, and economical use of antigen stocks. Each antigen (60 pmol/106 beads) was coupled to a separate region of MAGPLEX^®^ magnetic microspheres (Luminex Corporation, MV’s-Hertogenbosch, the Netherlands) using the xMAP^®^ Antibody Coupling Kit (Luminex Corporation, MV’s-Hertogenbosch, The Netherlands) following manufacturer’s recommendations. Samples diluted 1/400 in Wash Solution (Thermo Fisher Scientific, Waltham, MA, USA) were incubated with 1000 coupled beads for each antigen per well for 1 h at room temperature in a light-protected plate shaker. After washing, the beads were incubated with R-Phycoerythrin AffiniPure F (ab’)_2_ Fragment Goat Anti-Human IgG (H + L) (Jackson ImmunoResearch, West Grove, PA, USA) for 1 h at room temperature in a light-protected plate shaker. After washing, antigen-antibody reactions were read on a MAGPIX^®^ system using xPONENT^®^ Software (Luminex Corporation, MV’s-Hertogenbosch, the Netherlands) and the results were expressed as median fluorescence intensity (MFI). Using a panel of 486 plasma donors tested with commercial Anti-SARS-CoV-2 Spike S1-domain and -SARS-CoV-2 NCP IgG ELISAs from Euroimmun and a virus neutralization test in order to determine their serological status against SARS-CoV-2, the cut-off values and assay performance indicators were calculated by receiver operating characteristic curve (ROC) analysis. The specificity/sensitivity values for the CTD and NTD domains in the duplex assay were 96.1%/97.8% and 87.8%/88.5%, respectively.

A second test was performed to detect IgG antibodies against the S portion. According to the manufacturer’s (EUROIMMUN, Lübeck, Germany) instructions, a test was considered ELISA-S-positive when the results indicated an optical density ratio ≥ 1.1 (sensitivity, 87%; specificity, 97.5%) [17]. Antibody levels for ELISA-S were assigned an arbitrary binding unit (BAU) to allow comparison with other studies, as recommended by the seventy-fourth report (WHO TRS N°1039) of the WHO Expert Committee on Biological Standardization.

Finally, an in-house virus neutralization test (VNT) was used to detect neutralizing anti-SARS-CoV-2 antibodies [18]. Vero E6 cells were cultured in 96-well microplates with 100 50% tissue culture infective doses of the SARS-CoV-2 ancestral strain BavPat1 (courtesy of Prof. Drosten, Berlin, Germany) and serial dilutions of each serum sample (1/10–1/10,240). Dilutions associated with a cytopathic effect were considered negative (no neutralization), and those with no such effect at day 4 post-infection were considered positive (complete neutralization). The neutralization titer was defined as the highest dilution of serum that still yielded neutralization. Specimens with a VNT titer ≥ 20 were considered positive with a specificity of 100%. The participants received their serology test results by mail or e-mail.

### 2.5. Outcomes

The main outcome of this study was the presence of SARS-CoV-2 neutralizing antibodies defined as a titer ≥ 20. The secondary outcome measures were defined as the following: (i) a positive ELISA-S test, (ii) a positive ELISA-N test.

### 2.6. Variable Definitions

During the study period, the primary COVID-19 vaccination consisted of two vaccine doses, except for the vaccination of people who had recovered from COVID-19 within 6 months before vaccination [5,6]. Three doses were recommended for frail adults in August 2021 [19].

Participants were grouped into four different categories based on vaccination status and then self-reported biologically confirmed SARS-CoV-2 infection since January 2020. Not vaccinated participants represent participants who did not receive any doses of a COVID-19 vaccine. Vaccinated participants were grouped according to the number of COVID-19 vaccine doses (either one, two, or three doses). 

A self-reported confirmed SARS-CoV-2 infection was defined as having a positive SARS-CoV-2 reverse transcription-quantitative polymerase chain reaction (RT-qPCR), antigenic test, and/or enzyme-linked immunosorbent assay (ELISA) test result (before the first vaccine dose for vaccinees) since January 2020. Self-reported SARS-CoV-2 testing history was assessed using the following question: “Since January 2020, have you ever tested positive for a SARS-CoV-2 infection (RT-qPCR and/or antigenic test)?” (Responses: “Yes”, “No”, “Don’t Know”). This variable is cited in the text as “self-reported SARS-CoV-2 infection”. Historical strain was defined as having reported a confirmed SARS-CoV-2 infection before 31 December 2020, consistent with the virus circulation in France [11].

A previous SARS-CoV-2 infection was defined as a self-reported SARS-CoV-2 infection (as previously defined) and/or yielding a positive ELISA-N test result for the sera analyzed in this study. The variable “previous SARS-CoV-2 infection” has been used in the univariate and multivariate analysis (see §Statistical analysis).

Participants who reported a biologically confirmed SARS-CoV-2 infection more than twenty-one days after a complete primo-vaccination schedule were considered post-vaccination infected in concordance with recommendations. A complete primo-vaccination schedule was defined as having received two doses or one dose at least three months after a biologically confirmed SARS-CoV-2 infection in accordance with recommendations in France [5]. 

### 2.7. Statistical Analysis

The minimum sample size was initially calculated assuming the following: an a priori 10% seroprevalence of anti-SARS-CoV-2 IgG acquired by natural immunity (ELISA-S+ and ELISA-N+) among participants in each PHCW subpopulation, a confidence in the estimate of 95%, a maximum allowable error in the prevalence of 3%, and a minimum 20% dropout rate. Details regarding this aspect have been described elsewhere [16]. For the descriptive analysis, we selected all consenting participants who provided a serology test and returned the DBS before 31 August 2021, had interpretable serology results, and had completed the questionnaire. Standard descriptive statistics were used to present the sociodemographic and clinical characteristics of the participants. Categorical data are presented as counts and percentages. Data on continuous variables are presented as the median and inter-quartile range (IQR). 

Seropositivity of neutralizing and IgG antibodies against the SARS-CoV-2 N (ELISA-N) and S (ELISA-S) proteins and their 95% confidence intervals (CIs) were estimated for all PHCWs. The seropositivity in each assay was calculated using the rate of the number of participants with a positive antibody test as the numerator and the number of participants as the denominator. Seropositivity was also presented according to the number of doses received and the presence or absence of self-reported SARS-CoV-2 infection. To calculate the 95% confidence interval for seroprevalence, we used a normal approximation interval except for samples with a seroprevalence < 5% or >95% or for a sample size < 30, when the Clopper Pearson exact method based on binomial distribution was used [20]. ELISA-S titers (BAU/mL) showed a skewed distribution and were log transformed. Log-transformed antibody levels were compared using Student’s *t* test and analysis of variance between groups defined by the following: (i) number of COVID-19 vaccine doses (zero, one, or two); (ii) the presence or absence of self-reported SARS-CoV-2 infection; (iii) time since the last COVID-19 vaccination (months); (iv) age (<40 years, 40–49 years, 50–59 years, or ≥60 years. The Spearman rank test was used for the analysis of correlations between the quantitative levels of anti-SARS-CoV-2 S1 IgG and the titers of neutralizing antibodies. The results were interpreted as negligible (<0–0.19), weak (0.2–0.39), moderate (0.40–0.69), strong (0.70–0.89), or very strong (>0.9).

To identify the factors associated with the detection of neutralizing antibodies, we used a binary logistic regression model. All the following variables with *p*-values < 0.2 in the univariable analysis were considered in the multivariable analysis: age, sex, presence of chronic diseases, number of vaccine doses received, and previous SARS-CoV-2 infection. A step-by-step backward elimination procedure was used to identify the independent covariates associated with the presence of neutralizing antibodies in the multivariable analysis (*p*-value level of 0.05). To explore whether the association between the detection of neutralizing antibodies and a previous SARS-CoV-2 infection might differ according to the number of vaccine doses received, the interaction between the number of vaccine doses and previous SARS-CoV-2 infection was explored for effect modification. To evaluate the potential association between the time since vaccination and the presence of neutralizing antibodies, the variable of time since the last vaccination (in months) was also tested in models among the PHCW who received one dose and those who received two doses. Individuals with missing data for covariables considered in the regression models were excluded from the analysis. All statistical analyses were conducted using R software, version 4.0.3 (R Foundation, Vienna, Austria) [21]. Statistical significance was set at *p* < 0.05.

### 2.8. Ethics

This work was approved by the “CPP Île de France V” ethics committee (ID RCB: 2020-A03298-31). Written informed consent was obtained from all participants.

## 3. Results

### 3.1. Characteristics of Study Participants

From June 2021 to August 2021, a total of 1612 PHCWs (527 GPs, 430 pediatricians, 381 dentists and assistants, and 274 pharmacists and assistants) who completed a questionnaire and had an interpretable serological analysis were included in this study. A flow chart of the PHCW participants is shown in Figure 1. 

The majority of participants were women (69%; *n* = 1112) and the median participant age was 47 (range, 21–79) years (Table 1). 

Approximately 19.2% (*n* = 309) of the participants had a chronic disease. Of the 1612 participants, 15.7% (*n* = 253) self-reported a SARS-CoV-2 infection. Among the 191 PHCWs who reported a positive RT-qPCR and/or antigenic result, 185 (94.4%) declared having at least one symptom at the time of the diagnostic test. Regarding the COVID-19 vaccination schedule (vaccine doses and/or self-reported SARS-CoV-2 infection) of the 1604 PHCWs for whom the information was available, 80.5% (*n* = 1292) had received two vaccine doses, 11.6% (*n* = 186) one vaccine dose, and 0.1% (*n* = 2) three vaccine doses, whereas 7.8% (*n* = 124) declared that they had not been vaccinated (Table 1). Among the 1480 PHCWs who received at least one vaccine dose, 87.9% (*n* = 1301) were vaccinated using the BNT162b2 mRNA vaccine. The details of the vaccine types received by the participants are provided in Appendix A.

### 3.2. Characteristics of Participants with Self-Reported SARS-CoV-2 infection after Primary COVID-19 Vaccination

Of those with a primary COVID-19 vaccination, 0.3% (*n* = 4 out of 1417) reported having a positive diagnostic COVID-19 test at least 21 days after being vaccinated. Their characteristics are presented in Table 2.

The median age of this group was 57 (range, 51–63) years and 75% were men (*n* = 3). None declared having a biologically confirmed SARS-CoV-2 infection before vaccination. All experienced a symptomatic post vaccination infection; one was positive for the α variant, one for the γ variant, and the variant had not been identified for the other two participants. The time between the last vaccination and infection ranged from 1 to 3 months, and the time between infection and study sampling ranged from 2 to 3 months. All these participants showed seropositivity for IgG against the SARS-CoV-2 S and N proteins. At the time of the survey, neutralizing antibodies were detected among all of them.

### 3.3. Seroprevalence of SARS-CoV-2 Antibodies

Table 3 shows the proportion of participants with a positive ELISA result for IgG antibodies against the SARS-CoV-2 N and S proteins and positive neutralizing antibodies according to the vaccination schedule.

### 3.4. Seroprevalence of Antibodies against the SARS-CoV-2 N Protein

A total of 381 of the 1612 PHCWs analyzed tested positive for antibodies against the SARS-CoV-2 N protein for a prevalence rate of 23.6% (95% CI 21.6%–25.7%, Table 3). Among them, 47.2% (*n* = 180) had a self-reported SARS-CoV-2 infection. Among the 251 PHCWs who had a self-reported SARS-CoV-2 infection, 71.9% (*n* = 180) had a positive ELISA-N test, whereas among the 1353 PHCW not declaring a self-reported SARS-CoV-2 infection, 14.6% (*n* = 197) had a positive ELISA-N test (*p* < 0.0001). 

### 3.5. Seroprevalence of Antibodies against the SARS-CoV-2 S Protein and Quantitative ELISA-S Results

Among the 1612 PHCWs, 1526 tested positive for antibodies against the SARS-CoV-2 S protein for a prevalence rate of 94.7% (95% CI 93.6%–95.7%) (Table 3). The seropositivity rate for antibodies against the S protein was nearly 100% among the PHCWs who had received two vaccine doses or one vaccine dose and a self-reported SARS-CoV-2 infection. 

Among the 1526 PHCWs with detectable IgG antibodies against the S protein, the mean antibody level was 1168.81 BAU/mL. Comparisons of IgG antibody levels (BAU/mL) against the S protein with the vaccination and participant characteristics are presented in Figure 2.

The line in the center of each box depicts the median; the lower and upper hinges correspond to the first and third quartiles; and the distance between the first and third quartiles corresponds to the interquartile range. Student’s *t* test and analysis of variance were used to assess statistically significant differences in antibody levels according to the number of vaccine doses, the presence or absence of self-reported SARS-CoV-2 infection, the time since vaccination, and age categories.

A significantly higher mean antibody level was found among the PHCWs who had a self-reported SARS-CoV-2 infection than those who did not, except for unvaccinated PHCWs (zero vaccine dose, 297.95 BAU/mL versus 58.37 BAU/mL, *p* = 0.1434; one vaccine dose, 2089.69 BAU/mL versus 500.87 BAU/mL, *p* < 0.0001; two vaccine doses, 2298.07 BAU/mL versus 1076.21 BAU/mL, *p* < 0.0001; Figure 2a). Among the PHCWs without a self-reported SARS-CoV-2 infection, there was no statistically significant difference in the mean antibody level for those who received one vaccine dose compared to those who did not receive the vaccine, although it was close to significance (500.87 BAU/mL versus 58.37 BAU/mL; *p* = 0.0779). Among the PHCWs without a self-reported SARS-CoV-2 infection, there was a higher mean antibody level for those who received two vaccine doses than those who received one vaccine dose (1076.21 BAU/mL versus 500.87 BAU/mL; *p* < 0.0001). Among the PHCWs with a self-reported SARS-CoV-2 infection, a significantly higher mean antibody level was observed for those who received one vaccine dose than those who did not receive the vaccine (2089.69 BAU/mL versus 297.95 BAU/mL; *p* < 0.0001) but the antibody levels were similar for those who received two vaccine doses compared to those who received one vaccine dose (2298.07 BAU/mL versus 2089.69 BAU/mL; *p* = 0.2820). The quantitative ELISA-S results showed a decrease in antibody levels as the time since the last vaccination (*p* < 0.0001) or age (*p* < 0.0001) increased (Figure 2b,c, respectively). A significantly higher mean antibody level was observed for participants who received their last vaccine dose within the 3 months before participating in the study compared to those vaccinated more than 3 months before the study (*p* < 0.0001), and among those aged less than 50 years compared to those aged 50 years or more (*p* < 0.0001).

### 3.6. Seroneutralizing Antibodies and Correlation with ELISA Test

A total of 1311 PHCWs (81.3%; 95% CI 79.4–83.2%) had a positive result for the VNT (Table 3). The seropositivity rate for neutralizing antibodies against SARS-CoV-2 was >90% for the PHCWs who received at least one vaccine dose and had a self-reported SARS-CoV-2 infection. This rate was 85.4% for those who received two vaccine doses and did not have a self-reported SARS-CoV-2 infection (*n* = 1062 out of 1244), 62.9% for those who were not vaccinated but had a self-reported SARS-CoV-2 infection (*n* = 39 out of 62), 42.2% for those who received one vaccine dose without a self-reported SARS-CoV-2 infection (*n* = 19 out of 45), and 6.5% for those who were not vaccinated and did not have a self-reported SARS-CoV-2 infection (*n* = 4 out of 62; Table 3).

As shown in Figure 3, among the 1311 participants with neutralizing antibodies, 104 (7.9%), 295 (22.5%), 294 (22.4%), 252 (19.2%), 197 (15.0%), 112 (8.5%), and 57 (4.3%) participants had VNT titers of 20, 40, 80, 160, 320, 640, and ≥1280, respectively. 

The anti-S protein IgG titers were moderately correlated with the neutralizing antibody titers (Spearman’s rank correlation coefficient, 0.65) that was statistically significant (*p* < 0.0001; Figure 4).

### 3.7. Factors Associated with the Detection of Neutralizing Antibodies

Table 4 shows the results of univariable and multivariable analyses of factors associated with the detection of neutralizing antibodies among the PHCWs.

Because the interaction term between the number of COVID-19 vaccine doses and the presence or absence of previous SARS-CoV-2 infection was statistically significant (*p* < 0.0001), we have presented subgroup analyses according to the number of vaccine doses. Among the participants who were not vaccinated (*n* = 124) and those who received one vaccine dose (*n* = 186), multivariable regression analyses showed that detection of neutralizing antibodies was significantly more likely in those with previous SARS-CoV-2 infection than in those with no previous SARS-CoV-2 infection (zero vaccine doses, odds ratio (OR) 16.57, 95% CI 5.96–59.36; one vaccine dose, OR 41.66, 95% CI 16.05–120.78). Among participants who received two vaccine doses (*n* = 1292), multivariate regression analyses showed that the detection of neutralizing antibodies was not significantly associated with previous SARS-CoV-2 infection (OR 1.31, 95% CI 0.86–2.07), but was more likely for those who received their second vaccine dose within the 3 months before participating in the study than those vaccinated more than 3 months before the study (OR 5.28, 95% CI 3.51–8.23). In all multivariate regression analyses, age, sex, and chronic disease were not statistically associated with the detection of neutralizing antibodies.

## 4. Discussion

The findings of this cross-sectional nationwide study, which included serum samples from 1612 PHCWs collected from June 2021 to August 2021, demonstrated a stronger immune response in PHCWs with a previous SARS-CoV-2 infection than those with no evidence of a previous SARS-CoV-2 infection, particularly PHCWs with a single vaccine dose. This is in line with previous studies [22,23,24] reporting higher and longer-lasting anti-S antibody levels among individuals with evidence of a previous SARS-CoV-2 infection than vaccinated individuals, which represents a remarkable and sustained enhancement of both the humoral and cellular responses, including higher neutralizing antibody responses [25,26,27,28]. As in previous studies [26,27,28], we show that PHCWs who experienced COVID-19 after a single dose of vaccine exhibited antibody titers similar to those that had received two doses of vaccine. This is in contrast with the marked antibody increase after the second dose of vaccine in PHCWs with no evidence of a previous infection before vaccination. This striking difference in antibody kinetics between previously infected and vaccinated individuals could explain the significantly lower incidence of breakthrough infection in previously infected individuals than vaccinated individuals [29,30,31]. In agreement with clinical trials [2] and real-world settings [32,33], we observed that PHCWs who received a two vaccine doses had higher antibody responses than unvaccinated individuals or those who received a single vaccine dose and had no reported SARS-CoV-2 infection. We found that a second vaccine dose in pre-exposed PHCWs did not increase their antibody levels, in line with previous studies [23,34]. This supports the strategy of a single-dose vaccination for PHCWs who experienced COVID-19 to achieve a primary vaccination.

Not surprisingly, given the kinetics of immune responses, the degree of protection offered by a vaccine against infection may decrease over time, allowing for more breakthrough infections as the immune response wanes over months. Notably, we identified a significant correlation between time-from-vaccine and antibody levels by reporting waning of humoral responses more than 3 months after the administration of the second vaccine dose compared to the first 3 months. This result is consistent with recent publications showing a significant decline in S-antibody levels and neutralizing antibody levels with time after the second vaccine dose [35,36,37,38,39]. The clinical implications for the waning of the humoral response after vaccination are therefore still unclear and it remains crucial to establish the S antibody threshold associated with protection against clinical outcomes [35,36,37]. In our sample, four participants who had been vaccinated declared a SARS-CoV-2 infection post vaccination. Although we cannot draw robust statistical conclusions from so few cases, it is interesting to note that none of them had a previous SARS-CoV-2 infection, and that the time between last vaccination and infection was between 1 and 3 months. Notably, these estimates were obtained at times when the α variant was dominant, and the estimated titers may be less effective against infection with the δ variant [40]. Determining the immune correlates for protection against SARS-CoV-2 is necessary to predict how a decrease in antibodies will affect clinical outcomes. This will also help determine if and when a booster dose will be needed, and whether vaccinated individuals are protected from SARS-CoV-2. In line with previous studies, we found a correlation (albeit moderate) between anti-S and neutralizing antibodies [30]. An 80% vaccine efficacy against symptomatic SARS-CoV-2 infection was observed for an antibody level of 264 BAU/mL, which declined to 60% for a level of 54 BAU/mL [41]. Our findings suggest that antibody titers had declined below these threshold levels in a subgroup of the PHCWs within 3 months of the second vaccination, indicating the need for a booster. Although we did not find that antibodies waned over time in PHCWs who received one vaccine dose, vaccine recall should be considered when a long time has passed since natural infection or vaccination, because antibodies consistently decline over time. In the present study, almost one-third of the PHCWs had antibodies against the SARS-CoV-2 N protein that developed as the result of a natural SARS-CoV-2 infection. Importantly, anti-N IgG antibodies remained quite stable over a period of at least 3 months following mild or asymptomatic infection [42]. The results reported here mainly relate to the circulation of SARS-CoV-2 among the participants since December 2020 at the end of the second wave in France and at the start of the COVID-19 vaccination campaign. This finding indicates that PHCWs represent a population at considerable risk of contracting COVID-19 and highlights the relevance of vaccination for this population.

### Strengths and Limitations

The main strength of this real-world study was the use of three serological tests among PHCWs, which were one of the first populations to have access to COVID-19 vaccines because of their high risk of SARS-CoV-2 exposure. Another strength was the availability of sociodemographic and clinical data. Nevertheless, our study also had several limitations. First, we did not evaluate the cellular immune response, though a correlation between cellular and humoral immunity has been reported [43]. Second, it is possible that the responders in this study may not be representative of the population as a whole, therefore limiting generalizability of our findings. In particular, the HCW sample is composed mostly of young adults, and it is important to consider the elderly in studies on immune responses, since they may have a weaker antibody response to vaccination. Third, although we measured antibodies against the N protein of SARS-CoV-2, all previous asymptomatic cases were certainly not detected given the short duration of detectable N antibodies [44]. This may lead to misclassification in pre-exposure groups. In addition, we were not able to model the antibody response according to previous symptomatic versus asymptomatic SARS-CoV-2 infection. Fourth, although we controlled for age, sex, and chronic disease, our findings might have been affected by other possible confounders that were not assessed. Fifth, we relied on self-reported results for SARS-CoV-2 infection, which may have introduced a bias in the reporting of results, leading to misclassification. Other study limitations include insufficient data to model the antibody response according to different vaccines.

## 5. Conclusions

Immunity due to either previous SARS-CoV-2 infection or vaccination can contribute to herd immunity and control the COVID-19 pandemic. Our study, together with others, underscores the fact that previous SARS-CoV-2 infection should be considered when planning booster doses and the design of COVID-19 vaccination strategies. Further studies should be conducted in the following months and after the booster dose to reassess seroprevalence and identify factors associated with infection among PHCWs.

## Figures and Tables

**Figure 1 viruses-14-00957-f001:**
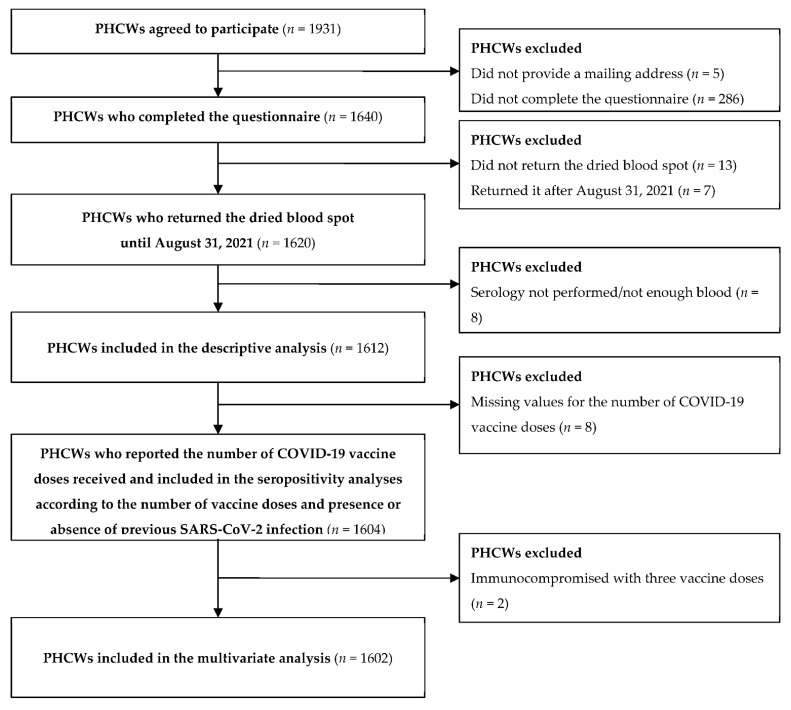
Flow chart of the primary healthcare worker (PHCW) participants enrolled from May to August 2021 (COVID-SéroPRIM study, France, 2021).

**Figure 2 viruses-14-00957-f002:**
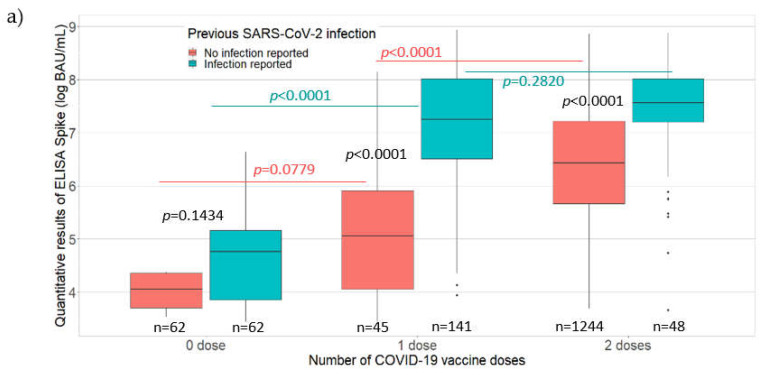
Antibody levels (BAU/mL) for IgG antibodies against the S protein according to (**a**) the number of COVID-19 vaccine doses received along with the presence or absence of self-reported SARS-CoV-2 infection among the PHCWs; (**b**) time since the last vaccination among vaccinees; (**c**) age (years) of the PHCWs, May 2021 to August 2021 (*n* = 1602; COVID-SéroPRIM study, France, 2021).

**Figure 3 viruses-14-00957-f003:**
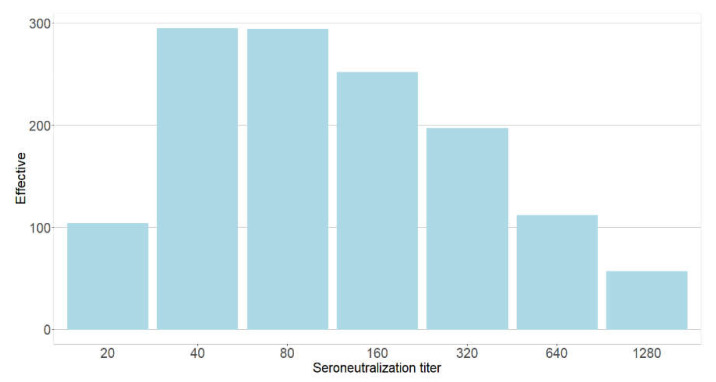
Distribution of seroneutralization titers among the PHCWs, May 2021 to August 2021 (*N* = 1602; COVID-SéroPRIM study, France, 2021).

**Figure 4 viruses-14-00957-f004:**
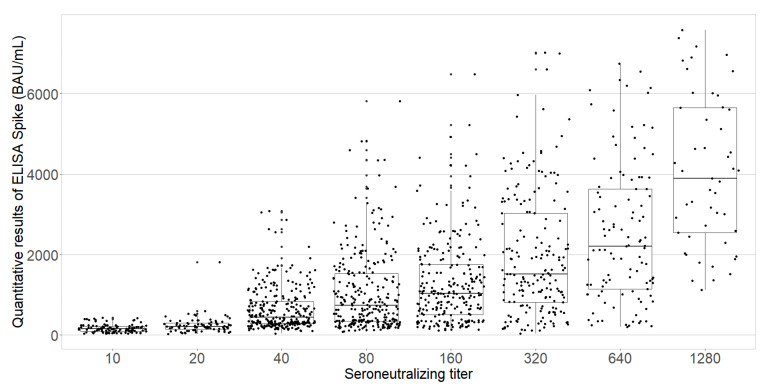
Boxplot of the quantitative ELISA-S results according to neutralizing antibody titers among the PHCWs (*N* = 1602; COVID-SéroPRIM study, France, 2021).

**Table 1 viruses-14-00957-t001:** Sociodemographic and clinical characteristics of the PHCWs (*N* = 1612; COVID-SéroPRIM study, France, 2021).

Variable	Total (*N* = 1612)	
	*n*	%
**Median age** (min–max)	47 (21–79)	
**Age group** (years)		
<40	430	26.7
40–49	460	28.5
50–59	420	26.1
≥60	302	18.7
**Sex**		
Female	1112	69.0
**Geographical area**		
Ile de France	297	18.4
Northeast	387	24.0
Southeast	362	22.5
Northwest	304	18.9
Southwest	262	16.3
**Occupation**		
GP	527	32.7
Pediatrician	430	26.7
Dentist	331	20.5
Dental assistant	50	3.1
Pharmacist	238	14.8
Pharmacist assistant	36	2.2
**Chronic disease**		
Yes	309	19.2
**Self-reported SARS-CoV-2 infection confirmed biologically * since January 2020**	253	15.7
RT-qPCR and/or antigenic confirmed	191	75.5
*Symptomatic infection* ^¥¥¥^	185	94.4
*Historical strain*^¥^ ELISA confirmed	138 62	72.3 3.8
SARS-CoV-2 infection ≤6 months ^¥¥^	126	50.2
**COVID-19 vaccination and/or reported SARS-CoV-2 infection * (*N* = 1604)**		
**Not vaccinated**	**124**	**7.8**
No self-reported SARS-CoV-2 infection	62	3.9
Self-reported SARS-CoV-2 infection	62	3.9
**One vaccine dose**	**186**	**11.6**
No self-reported SARS-CoV-2 infection	45	2.8
Self-reported SARS-CoV-2 infection	141	8.8
**Two vaccine doses**	**1292**	**80.5**
No self-reported SARS-CoV-2 infection	1244	77.6
Self-reported SARS-CoV-2 infection	48	3.0
**Three vaccine doses without self-reported SARS-CoV-2 infection ****	**2**	**0.1**

* Self-reported SARS-CoV-2 infection was defined as a positive RT-qPCR or antigenic and/or ELISA test declared by the participants since January 2020. ^¥^ Historical strain was defined as being tested RT-qPCR positive before 31 December 2020, consistent with the virus circulation in France. ^¥¥^ Information available for 251 participants. ^¥¥¥^ Information available for 190 participants. ** Participants who received three vaccine doses were considered immunocompromised, in accordance with the COVID-19 vaccination recommendations at the time of the survey.

**Table 2 viruses-14-00957-t002:** Characteristics of the PHCWs who tested positive for SARS-CoV-2 after a complete primary COVID-19 vaccination schedule (*N* = 4; COVID-SéroPRIM study, France, 2021).

	Sociodemographic Characteristics			Vaccination			Infection					Seropositivity		
PHCW	Age (Years)	Sex	Occupation	Number of Doses	Vaccine (Name First Vaccine Dose* Name Second Vaccine Dose)	Time between Vaccination (Last Injection) and Study Sampling (Months)	Before Vaccination	After Vaccination: Variant	Time between Vaccination and Infection (Months)	Time between Infection and Study Sampling (Months)	Symptoms	Spike	Nucleocapsid	Neutralizing Antibodies (VNT Titer)
1	63	Male	GP	2	BNT162b2*missing value	5	No	Not known	2	3	Rhinorrhea	Pos	Pos	1280
2	60	Male	Dentist	2	BNT162b2* BNT162b2	4	No	Alpha	2	2	Cough, rhinorrhea, sore throat, diarrhea, chest pain or tightness, loss of smell, sleep disorders	Pos	Pos	160
3	51	Male	Pediatrician	2	BNT162b2* BNT162b2	5	No	Gamma	3	2	Cough, fever, rhinorrhea, sore throat, headaches, chest pain or tightness, loss of smell, loss of taste, loss of weight, loss of appetite, tiredness, muscular pains, thrills	Pos	Pos	320
4	54	Female	Pediatrician	2	BNT162b2* BNT162b2	4	No	Not known	1	3	Cough, fever, tiredness, diarrhea, loss of smell, loss of taste, dyspnea while exercising, loss of appetite, muscular pains, heart rhythm disorders, dyspnea while performing activities of daily living, feeling of dizziness, loss of balance	Pos	Pos	640

A complete primo-vaccination schedule was defined as having received two doses or one dose at least three months after a biologically confirmed SARS-CoV-2 infection in accordance with recommendations in France [5].

**Table 3 viruses-14-00957-t003:** Proportion of participants with a positive ELISA result for IgG antibodies against the SARS-CoV-2 N and S proteins and positive neutralizing antibodies according to the number of vaccine doses received and the presence or absence of a self-reported SARS-CoV-2 infection, May 2021 to August 2021, (*N* = 1612; COVID-SéroPRIM study, France, 2021).

Variable	Seroprevalence					
	Anti-SARS-CoV-2 N Protein Antibodies		Anti-SARS-CoV-2 S Protein Antibodies		Seroneutralization	
	*N*	% (95% CI)	*N*	% (95% CI)	*N*	% (95% CI)
**Total (*N* = 1612)**	381	23.6 (21.6–25.7)	1526	94.7 (93.6–95.7)	1311	81.3 (79.4–83.2)
**COVID-19 vaccination and/or self-reported SARS-CoV-2 infection * (*N* = 1604)**						
**Not vaccinated**						
No self-reported SARS-CoV-2 infection (*N* = 62)	7	11.3 (3.4–19.2)	4	6.5 (0.3–12.6)	4	6.5 (0.3–12.6)
Self-reported SARS-CoV-2 infection (*N* = 62)	52	83.9 (74.7–93.0)	49	79.0 (68.9–89.2)	39	62.9 (50.9–74.9)
**One vaccine dose**						
No self-reported SARS-CoV-2 infection (*N* = 45)	8	17.8 (6.6–28.9)	34	75.6 (63.0–88.1)	19	42.2 (27.8–56.7)
Self-reported SARS-CoV-2 infection (*N* = 141)	97	68.8 (61.1–76.4)	139	98.6 (95.0–99.8)	133	94.3 (90.5–98.1)
**Two vaccine doses**						
No self-reported SARS-CoV-2 infection (*N* = 1244)	182	14.6 (12.7–16.6)	1242	99.8 (99.4–100.0)	1062	85.4 (83.4–87.3)
Self-reported SARS-CoV-2 infection (*N* = 48)	31	64.6 (51.1–78.1)	48	100.0 (92.6–100.0)	45	93.8 (86.9–100.0)
**Three vaccine doses**						
No self-reported SARS-CoV-2 infection (*N* = 2)	0	0	2	100.0 (15.8–100.0)	2	100.0 (15.8-100.0)

The normal approximation interval was used to calculate the 95% confidence interval for seroprevalence, except when the Clopper Pearson exact method based on binomial distribution was used for samples with a seroprevalence < 5% or >95% or for a sample size < 30 [20].

**Table 4 viruses-14-00957-t004:** Univariate and multivariate analyses of factors associated with the detection of neutralizing antibodies among the PHCWs according to the number of COVID-19 vaccine doses (*n* = 1602; COVID-SéroPRIM study, France, 2021).

Variables *	*n* (% Positive in VNT)	Univariate			Multivariate		
		Odds Ratio	95% CI	*p*-Value	Odds Ratio	95% CI	*p*-Value
**Zero vaccine dose**	**124**						
Age (years)				0.3759			
<50	73 (31.5)	Reference					
≥50	51 (39.2)	1.40	(0.66–2.97)				
Sex				0.9541			
Female	89 (34.8)	Reference					
Male	35 (34.3)	0.98	(0.42–2.20)				
Chronic disease				0.0555			
No	101 (30.7)	Reference					
Yes	13 (52.2)	2.46	(0.98–6.28)				
Previous SARS-CoV-2 infection **				**<0.0001**			**<0.0001**
No	55 (92.7)	Reference			Reference		
Yes	69 (56.5)	16.57	(5.96–59.36)		16.57	(5.96–59.36)	
**One vaccine dose**	186						
Age (years)				0.9499			
<50	114 (81.6)	Reference					
≥50	72 (81.9)	1.02	(0.48–2.25)				
Sex				0.1879			
Female	137 (79.6)	Reference					
Male	49 (87.8)	1.84	(0.75–5.20)				
Chronic disease				0.7136			
No	149 (81.2)	Reference					
Yes	37 (83.8)	1.2	(0.48–3.42)				
Previous SARS-CoV-2 infection **				**<0.0001**			**<0.0001**
No	35 (29.7)	Reference			Reference		
Yes	149 (94.6)	41.66	(16.05–120.78)		41.66	(16.05–120.78)	
Time since vaccination (months)				0.2641			
<3 ≥3	109 (78.9) 75 (85.3)	0.64	(0.28–1.39) Reference				
**Two vaccine doses**	1292						
Age (years)				**<0.0001**			
<50	701 (91.3)	Reference					
≥50	591 (79.0)	0.36	(0.26–0.50)				
Sex				**0.0385**			
Female	882 (87.1)	Reference					
Male	410 (82.7)	0.71	(0.51–0.98)				
Chronic disease				**0.0830**			
No	1046 (86.5)	Reference					
Yes	246 (82.1)	0.72	(0.50–1.05)				
Previous SARS-CoV-2 infection **				0.2082			
No	1062 (85.1)	Reference					
Yes	230 (88.3)	1.31	(0.86–2.07)				
Time since vaccination (months)				**<0.0001**			**<0.0001**
<3 ≥3	549 (95.1) 737 (78.6)	5.28	(3.51–8.23) Reference		5.28	(3.51–8.23) Reference	

* In a model including all participants, the interaction term between the number of COVID-19 vaccine doses received and the presence or absence of previous SARS-CoV-2 infection was statistically significant (*p* < 0.0001); thus, analyses using subgroups defined by the number of vaccine doses received were performed. ** Previous SARS-CoV-2 infection was defined as a self-reported SARS-CoV-2 infection and/or yielding ELISA-N test result achieved on the sera analyzed in this study.

## Data Availability

Not applicable.

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
