# Peer review of "Seroprevalence of SARS-CoV-2 IgG Antibodies and Factors Associated with SARS-CoV-2 IgG Neutralizing Activity among Primary Health Care Workers 6 Months after Vaccination Rollout in France"

_viruses, 2022, doi:10.3390/v14050957_

Round 1
Reviewer 1 Report
Decarreaux et al describe a cross sectional serosurvey for SARS-CoV-2 in primary health care workers in summer of 2021. The evaluate how demographics, prior COVID-19 disease, vaccine doses and time influence their results. This is a well written and informative paper. One strength is the use of N, S1 serology and neut assays. I do however have some suggestions for improvement, mostly related to transparency on methods.
- There is no description of how dried blood spots are processed to extract "serum". The use of serum is noted for the assays and the S serology data is then converted to WHO IU. Without a description of these methods and a defined method to infer blood volume from a dried blood spot, the magnitude of the reported results are in question.
- There is no description of how the N based luminex assay was designed, performed or analyzed- antigen source? validation? cutoff values etc.
- Need to state which WHO IS was used (ie reference number) as there are more than one.
- Does the information in Table 1 (ie the study population) accurately represent PHCW in general? Ie a comparison between study groups and known PHCW demographics? If not, this could introduce bias and should be mentioned as a limitation.
- Table 1 "Self-reported...." is confusing. Do you mean that of the 253 that only 191 had RTPCR and/or Ag confirmation and that the rest had ELISA confirmation? If so, i would list the ELISA confirmation group as well so that the two can add up. Also, am i to understand that you also mean that of the 191 who were confirmed, 185 were symptomatic or does that refer back to the total 253?
- There is no explanation of what "historical strain" means in this table. Please clarify.
- Table 2 PHCW #1- how is it that this person received two doses but the vaccines listed are BNT and N/A. I assume NA to mean Not applicable, so therefore they only received 1 dose?
- Table 2 Time between infection and sampling i also assume to be months? This is not noted in this column.
- Figure 2 graphs are way too small and Y axis labels are either not there or unreadable even if i use the zoom function on the PDF. Fix this. Note to authors: Graphpad Prism makes very lovely easy to adjust graphs, and i have no COI related to this recommendation.
- Same for Figure as #9.
- As a PHCW myself, what i really want to know is if the different groups of PHCW had different rates of prior infection (by self-report or N serology) as might be associated with exposure risk- ie the dentists and pediatricians maybe higher risk than the GP and the pharmacists given regular exposure to unmasked patients. Were there other serologic differences (neut titers? S1 titers?) between these groups of PHCW?
- Please note what WHO group the strain used for neut assays falls into (ancestral, alpha, delta etc)
Reviewer 2 Report
The work is devoted to an important and interesting study - the assessment of the dynamics of the immune response in different groups of individuals (vaccinated, recovered and vaccinated) and the relationship between the level of antigen-specific IgG and the level of virus-neutralizing antibodies.
As the manuscript was read, a number of comments and questions arose, as follows:
- When collecting blood, the authors used dried blood spot. – how was the serum collected from these spots? The question also arises, how was the amount of collected material normalized to the volume of serum? For the analysis of the titer of neutralizing antibodies, the authors indicate that they titrated the serum, however, it is indicated above that the spots were collected.
- Paragraph 3.2 of the article: this paragraph describes 4 volunteers who fell ill after vaccination. However, Table 1 indicates that there were more such volunteers. It is necessary in this part of the manuscript to describe in more detail why these particular volunteers are described in such detail. Table 2, it seems to me, is better to be removed in the supplementary.
- Figures 2 and 3 - unfortunately, the figures are of very poor quality. It is impossible to see the captions on the figures. It is necessary to provide a drawing in a different format and quality. To date, it is not possible to analyze the presented data.
Unfortunately, due to the low quality of the provided graphic material, there is no possibility of an adequate assessment of the manuscript. The manuscript may be revised after the submission of high-resolution graphic materials, where all small details and signatures are visible.
Due to the impossibility of evaluating the study data, it is impossible to assess the adequacy of the conclusion.
Author Response
Please see attachement

Reviewer 3 Report
This paper deals with an interesting topic. Globally, I judge this paper a good job, but I have some remarks especially regarding statistical methods:
rows 209-211: the authors state "The seropositivity in each assay was calculated using the rates of total number of positive antibody tests as the numerator and the total number of antibody tests as the denominator". I wonder if this means that some health workers provided more than one test. Please, specify.
row 214: please, specify why the authors have log-transformed; moreover, have the authors verified the parametric assumptions before applying Student's t test and Anova?
row 216: The authors state that one group was defined as follows: "(i) number of COVID-19 vaccine doses (zero, one, or two) and the presence or absence of self-reported SARS-CoV-2 infection". I wonder how the authors have defined the combination between the number of doses and the self-reported infection (for instance, adding them?) or separating the two different conditions? In this case, I suggest to renumber as follows: "(i) number of COVID-19 vaccine doses (zero, one, or two); (ii) the presence or absence of self-reported SARS-CoV-2 infection; (iii)" and so on
Table 3: I think it's relevant to indicate which formula for the 95%CI for proportions has been used; in the case of percentages close either to 0% or to 100%, an asymmetrical CI was expected (using the correct method of Miettinen, for instance)
Fig 2b and 2c: It's not clear, in my opinion, to which comparison the p-value (p<0.0001) is referred. If, as I think, it is referred to global Anova, I suggest reporting also the p-values for multiple comparisons (at least for the significant ones)
Author Response
Please see attachement
